Subject Area:
cellular biology/developmental biology/ neuroscience

Keywords:
dendrite, da neuron, PVD, epidermis, pruning, L1CAM

Author for correspondence:
Wei-Kang Yang
e-mail: weikan@gate.sinica.edu.tw

A contribution to the special collection commemorating the 90th anniversary of Academia Sinica.

# Beyond being innervated: the epidermis actively shapes sensory dendritic patterning

Wei-Kang Yang and Cheng-Ting Chien

Institute of Molecular Biology, Academia Sinica, Taipei 115, Taiwan

  W-KY, 0000-0002-0000-260X; C-TC, 0000-0002-7906-7173

Sensing environmental cues requires well-built neuronal circuits linked to the body surface. Sensory neurons generate dendrites to innervate surface epithelium, thereby making it the largest sensory organ in the body. Previous studies have illustrated that neuronal type, physiological function and branching patterns are determined by intrinsic factors. Perhaps for effective sensation or protection, sensory dendrites bind to or are surrounded by the substrate epidermis. Recent studies have shed light on the mechanisms by which dendrites interact with their substrates. These interactions suggest that substrates can regulate dendrite guidance, arborization and degeneration. In this review, we focus on recent studies of *Drosophila* and *Caenorhabditis elegans* that demonstrate how epidermal cells can regulate dendrites in several aspects.

## 1. Introduction

Sensing environmental information, such as mechanical, chemical or thermal stimulation, is essential for animal survival and fitness. Over the course of evolution, metazoans have developed extremely specialized sensory organs to accomplish vision, hearing, taste, touch and smell. Noxious stimuli of high-temperature or mechanical type are detected via the skin, a sensory organ comprising epidermal cells that is conserved from nematodes to human. Sensory neurons that detect nociceptive stimulations extend highly branched dendrites to fully innervate the epidermis, making the skin the largest sensory organ [1,2]. Although the skin has been known as a sensory organ for a long time, systematic studies of how sensory neurons generate dendrites only started in invertebrate systems in the 1990s [3]. In a little over a decade, researchers had uncovered molecular and cellular mechanisms intrinsic to dendritic arborization and began to reveal direct interactions between sensory dendrites and epidermal cells. To date, mainly through studies of *Drosophila*, *Caenorhabditis elegans* and the vertebrate zebrafish, it has been demonstrated that the epidermis is not just passively innervated by dendrites, but in fact it actively instructs dendritic branching. In this review, we summarize and discuss recent progress in dendrite–epidermis interactions in the fruit fly and nematode systems. For recent findings in zebrafish, refer to the outstanding review [4].

The dendritic arborization (da) neurons of the peripheral sensory system in *Drosophila* larvae are a model system used to study dendritic arborization. The diversity and complexity of *Drosophila* da neurons are ideal for studying neuronal fate determination, dendrite–dendrite interactions and dendrite–substrate interplay [2,5]. These da neurons are classified into four classes (I–IV) based on increasing complexity of their dendrites [5]. The diversity of dendritic patterning often reflects distinct functions and requirements for receptive field sizes. Class I da neurons function as proprioceptors, mainly projecting primary branches and only a few side branches [6,7]. The dendritic branches of class I da neurons are enriched with microtubules [8], rendering them suitable for detecting the contraction wave during larval crawling. Class III da neurons

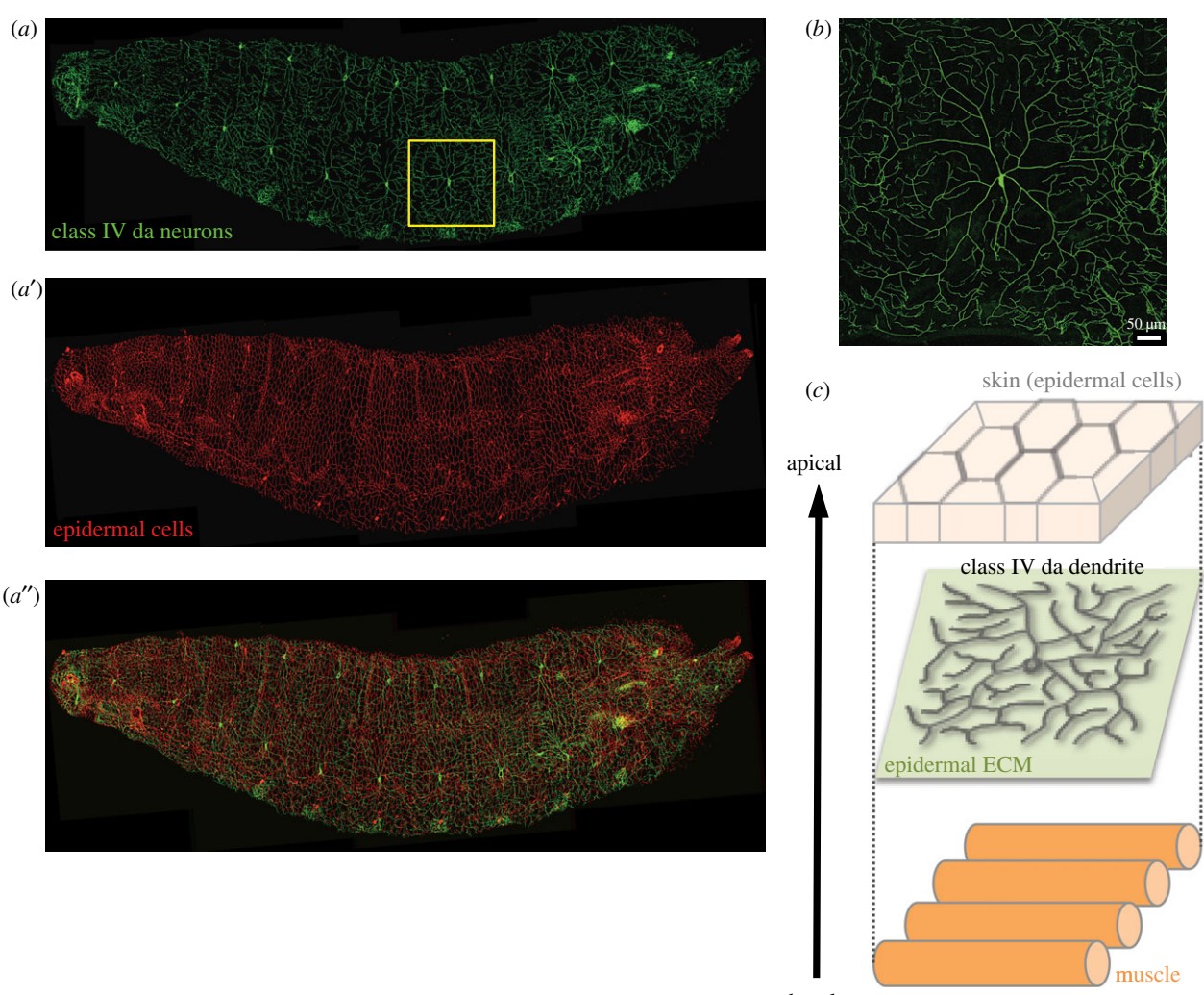

**Figure 1.** Class IV da neurons and epidermal cells in *Drosophila* larvae. A *Drosophila* larva co-expressing membrane-tethered GFP (mCD8-GFP) in class IV da neurons and membrane-tethered Tomato (CD4-tdTom) in epidermal cells. (*a*) GFP, (*a′*) tdTom and (*a″*) merged image. The yellow box indicates a single class IV da neuron and its receptive field. (*b*) A class IV da neuron located in the dorsal region of a larva, showing the complex but non-overlapping dendritic pattern. (*c*) Schematic of the epidermis – dendrite – muscle spatial arrangement. Dendrites are attached to the ECM (green plane) secreted by epidermal cells and are separated from muscles.

extend several long primary or secondary branches that protrude numerous spike-like branches, tiling 70–80% of the innervating field [5]. A recent study indicates that these spiked neurons function in sensing gentle touch via low-threshold mechano-transduction NOMPC channels located on the spikes [9]. The dendritic spikes are thin and built primarily of actin without microtubules, possibly making them more flexible and facilitating easier detection of weak mechanical force. Notably, animal body surfaces are filled with nociceptive sensors to instantly detect strong thermal stress and intense light from all directions so that they can escape these harmful stimuli. In *Drosophila* larvae, the class IV da neurons are capable of sensing multiple noxious stimuli [5,10]. Only three class IV da neurons innervate the epidermis of each hemi-segment, forming space-filling patterns and tiling each other (figures 1*a* and 2*b*) [5]. These complex dendritic patterns necessitate generating seventh- or eighth-order branches to reach up to 800 terminal ends for each neuron and to cover almost 100% of each hemi-segment [5]. These comprehensive dendritic branches express the mechano-transducers Painless [11], Pickpocket [12] and Piezo [13] for nociception and express the light sensor Gr28b [10] to detect strong light. Although class IV dendrites exhibit much more extensive branching and cover a larger

area than class III dendrites, these arbors are not responsive to gentle touch [9]. This functional limitation is primarily due to the lack of NOMPC expression. The mixed contents of actin and microtubules [14] could also possibly make class IV dendrites less flexible than class III dendritic spikes but suitable for high-threshold mechano-sensation. Therefore, elaboration of appropriate dendritic arbors and expression of correct sensory molecules are important for neuronal function. Based on genetic studies in *Drosophila* over the past two decades, key intrinsic regulators in dendritic patterning have been characterized [2]. Transcription factors function in concert to determine neuronal fate [8,15–18], cytoskeletal regulators and motor proteins shape dendritic patterns [19–23], secretory and endocytic pathways promote dendrite growth [24,25] and cell adhesion molecules (CAMs) limit dendrite outgrowth into the surroundings [26–29] and mediate dendritic self-avoidance [30–32].

The model organism *C. elegans* has emerged as an elegant system for studying dendrite–environment interactions, given the availability of powerful genetic tools. The two PVD neurons, each located on the left or right sides of this nematode, generate stereotypic menorah-like dendrites (figure 2*a*) [33]. Dendrite growth of PVD neurons is initiated by extending two first-order (1°) branches in the anterior and

royalsocietypublishing.org/journal/rsob    Open Biol. 9: 180257

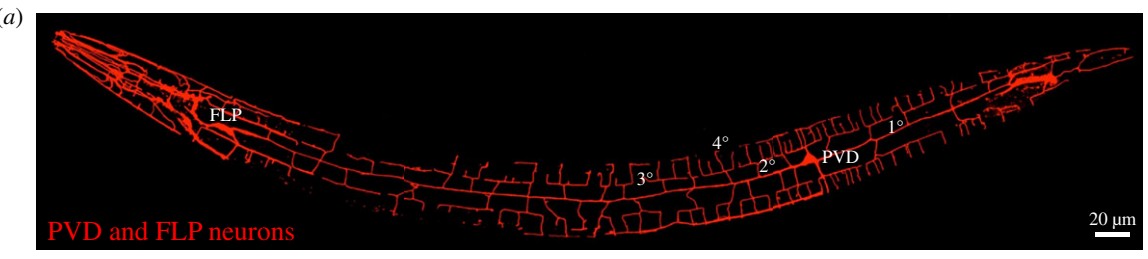

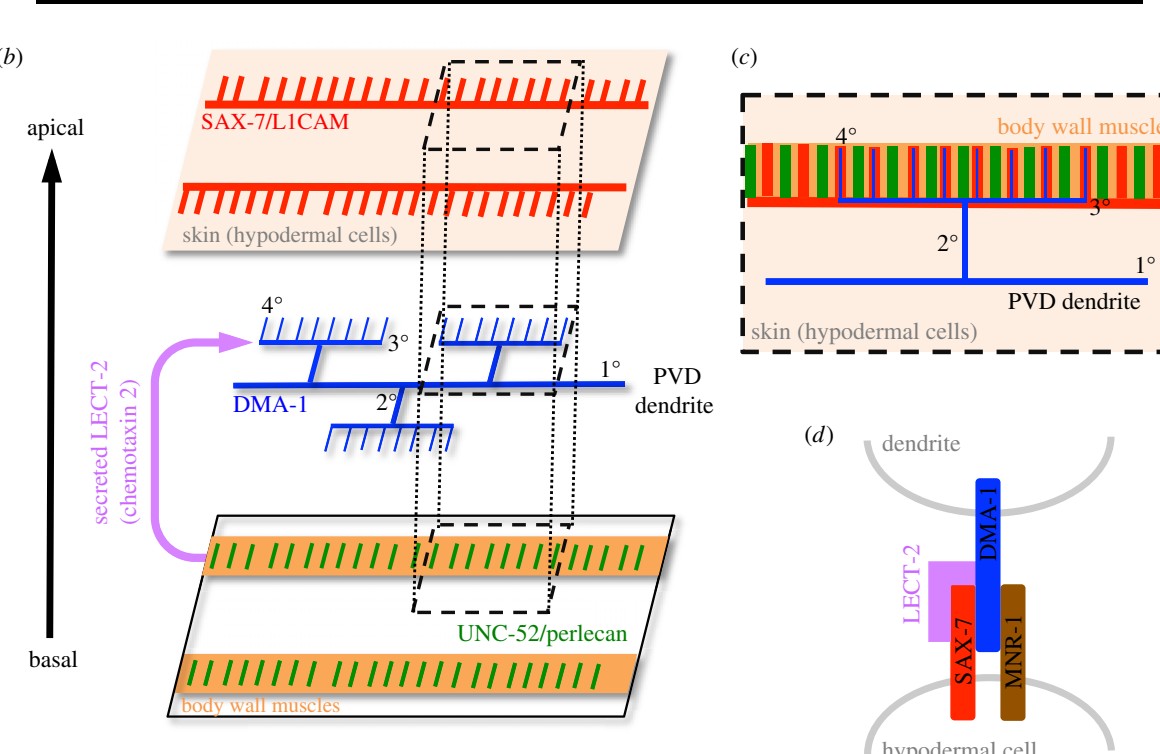

**Figure 2.** PVD neuron in *C. elegans*. (*a*) Somas and dendritic patterns of PVD and FLP neurons in a *C. elegans* larva are labelled by expressing membrane-tethered mCherry (myr::mCherry) in both neurons. Primary, secondary, tertiary and quaternary branches are indicated as 1°, 2°, 3° and 4° on PVD dendrites, respectively. FLP neuron and its dendrites are located in the anterior part, which are not overlapped to PVD dendrites. (*b*) Expression of guiding ligands and receptors in the interphase of skin, dendrites and muscle cells. SAX-7 (red) is expressed as horizontal stripes (corresponding to the 3° dendritic branches) and vertical stripes (corresponding to the 4° dendritic branches) on skin. DMA-1 (blue) is expressed on PVD dendrites. UNC-52 (green) is expressed as spaced stripes on the body wall muscles (orange). LECT-2 (magenta) is secreted from the body wall muscles to the dendrite–SAX-7 interface. (*c*) Dorsal view of menorah-like dendrites in the skin–dendrite–muscle arrangement represented by a grey dashed box in (*b*). Note that the 4° dendritic branches overlap with body wall muscle fibres (orange). (*d*) Receptor–ligand interactions between dendrites and skin.

posterior directions at larval stage 2 (L2). Then, at late L2/early L3, 2° branches sprout vertically from the 1° branches, extending to reach the lateral boundaries of outer body wall muscles. At the muscle boundary, most 2° branches bifurcate and extend horizontally to form 3° branches. Finally, at early L4, an array of short 4° branches extend vertically from the horizontal 3° branches to form menorah-like structures. To construct these highly branched dendrites, PVD neurons have been reported to use some intrinsic branching mechanisms that are analogous to those of *Drosophila* da neurons. For example, different levels of the transcription factor Cut confer class-specific da neuronal types in *Drosophila* [15], whereas low and high levels of the transcription factor MEC-3 confer, respectively, highly branching PVD neurons and restricted-branching AVM neurons in *C. elegans* [34,35]. Both systems are also analogous in generating distal branches. The dynein motor that functions in transporting cargo along microtubules is selectively required to promote distal branch formation in class IV da neurons [21]. Another study in *C. elegans* also identified mutations in the dynein accessory factor that selectively

disrupted distal branching of PVD neurons [36]. Thus, both invertebrate systems exhibit shared characteristics for some of the intrinsic mechanisms that promote dendrite branching.

Although it might be expected that both systems would also share some common mechanisms in terms of dendrite–epidermis interaction, which is the focus of this review, significant differences might also be anticipated due to differences in their dendritic morphologies and stereotypies, as well as their interacting environments. Functioning as nociceptors, dendrites of both class IV da and PVD neurons innervate a confined two-dimensional space between the epidermis and muscle cells (figures 1*c* and 2*b*). Whereas *Drosophila* class IV da neurons co-innervate the same field as other classes of da neurons, *C. elegans* PVD neurons exclusively target most body parts, leaving only the anterior segment to be innervated by the multi-dendritic FLP neuron (figure 2*a*). In addition, dendrites of *Drosophila* class IV da neurons are pruned completely in the pupal stage during metamorphosis and regrow new arbors for adulthood, which is not the case for *C. elegans* PVD neurons and thus prevents studies of remodelling in the nematode

system. However, the major difference between these two neuron types is the morphological complexity of their dendrites. Class IV da neurons elaborate numerous branches in a space-filling pattern to fully cover the target field (figure 1a,b), in a pattern similar to zebrafish Rohon–Beard (RB) sensory neurons [37] and retinal ganglion cells in mice [38]. These radial and extensively branching patterns are indicative of a space-filling mechanism whereby branches sprout and extend into uncovered areas. This mechanism has to be combined with a self-avoidance mechanism by which growth of branches is repulsed by iso-neuronal branches, preventing crossovers between branches (figure 1b) [39]. By contrast, C. elegans PVD dendrites are highly stereotypic, branching orthogonally from previous orders (figure 2a), so it would not be surprising if guidance cues are identified as being located along the growth pathways to direct this dendritic pattern. Thus, studies on these two systems provide distinct and complementary insights into how dendrites interact with epidermis during growth and patterning. Below, we first discuss the molecular and cellular mechanisms responsible for guiding dendritic branching of PVD neurons. We then focus on how epidermis regulates dendritic patterning of class IV da neurons.

## 2. Pre-patterning growth pathways

A novel transmembrane leucine-rich repeat (LRR) protein, DMA-1, is expressed in PVD neurons, and mutations in dma-1 disrupt higher-order dendrite branching after normal growth of 1° branches [40]. The LRR domain is located in the extracellular region of DMA-1 and mediates protein–protein interactions [41], suggesting the existence of unidentified extracellular binding partners to direct dendrite growth. Through forward genetic screens, two independent research groups found that PVD neurons in sax-7 mutants exhibit higher-order dendrite patterning defects; whereas 1° branches extend normally and some 2° branches successfully reach the sublateral muscle boundary, 3° branches fail to branch horizontally [42,43]. The sax-7 gene encodes an L1-type CAM of the evolutionarily conserved immunoglobulin CAM protein superfamily (IgSF CAM) [44]. As a transmembrane protein, L1CAM contains immunoglobulin domains linked to fibronectin III domains in the extracellular region, a single-pass transmembrane domain and a conserved cytoplasmic tail. In Drosophila and vertebrates, L1CAMs have critical functions in neurite outgrowth and guidance, and are expressed at high levels in neural tissues [25,45,46]. Interestingly, SAX-7 is expressed in non-neuronal hypodermal cells as sublateral longitudinal stripes that match the horizontal 3° branching pattern [42] (figure 2b,c). These results suggest that skin (hypodermal cells) directs pre-patterned growth of 3° branches. Indeed, ectopic expression of SAX-7 in lateral hypodermal seam cells of the sax-7 mutant, which are located along 1° branches, induced dendritic growth into the seam cells. Like other IgSF CAMs, skin-derived SAX-7 might interact with proteins presented on the dendritic surface via their extracellular domains. Through an additional genetic screen, mnr-1 mutations were identified as producing branching defects in PVD 3° branches similar to those displayed by dma-1 and sax-7 mutants [40,43]. The mnr-1 gene encodes an uncharacterized type I transmembrane protein [43]. Analyses of gene expression and mutant rescue suggest that

DMA-1 on dendritic surfaces interacts with SAX-7 and MNR-1 on hypodermal cell membranes. Drosophila S2 cells transfected with DMA-1 form strong aggregations with cells expressing both SAX-7 and MNR-1, but not with cells expressing only one or the other [42]. Accordingly, dendritic DMA-1 forms a tripartite complex with hypodermal SAX-7 and MNR-1 (figure 2d). Thus, skin cells appear to play a pre-patterning role by guiding dendrite branching via expression of SAX-7 in narrow stripes [47] (figure 3a). An immediate and obvious question is how SAX-7 expression is restricted to continuous stripes through skin. Recent forward genetic screens have shown that mutations in the dynein light intermediate chain DLI-1 and the cell–cell fusion protein EFF-1 caused disruption of PVD dendritic branching patterns similar to those observed in the sax-7 mutant [48]. Importantly, orderly SAX-7 stripes are scrambled into numerous ectopic patches between lateral and sublateral lines in both dli-1 and eff-1 mutants. Moreover, hypodermal but not neuronal expression of DLI-1 or EFF-1 rescues the respective mutant phenotypes in terms of SAX-7 stripe formation and PVD dendrite organization. Therefore, the microtubule-based transport system might be important for establishing or maintaining striped expression of SAX-7 in fused hypodermal cells.

The orderly array of PVD 4° branches hints at a pre-patterning event responsible for their parallel growth. Highly sensitive spinning-disk confocal microscopy revealed a stripe-like expression pattern of SAX-7 in skin arrays matching PVD 4° branching patterns, albeit weaker than that observed for 3° branches [49] (figure 2b,c). These SAX-7 stripes are expressed in hypodermal cells at the L3 stage, prior to the appearance of 4° branches, suggesting that terminal 4° branches might adopt the same strategy as 3° branches to grow along pre-patterned SAX-7 stripes. However, it is not possible to assay 4° dendrite branching defects in sax-7 mutants as their 3° branches are disrupted. Consequently, a genetic screen was undertaken for mutants in which only 4° branches are affected. This analysis identified mutations in unc-52, which encodes the extracellular heparan sulfate proteoglycan (HSPG) Perlecan, as specifically disrupting 4° branches, while 1°, 2° and 3° branches remained intact. This branch specificity largely depends on regulation of SAX-7 expression, as SAX-7 expression is normal in the 3° branching pathways but is absent from the 4° branching pathways in unc-52 mutants. Immunostaining revealed that UNC-52 is expressed in muscles, forming interdigitating stripes to SAX-7 stripes in the epidermis (figure 2b,c). Together with genetic studies, these findings resulted in a proposed model whereby localized UNC-52 stripes link muscle sarcomeres to hypodermal hemidesmosomes prior to SAX-7 stripe formation [49]. Thus, to make an array of well-spaced terminal branches, hypodermal substrates are first divided into equal units through local interactions at muscle contact sites, which then restricts SAX-7 stripe formation within these units (figure 3a). Interestingly, according to different orders of branches and local environments, extracellular matrix (ECM) proteins may shape dendrites in different ways. In the unc-52 mutant, apart from the effect on 4° branches, 2° branches are also reduced, although not with complete penetrance [50]. Away from the epidermis, the ECM protein NID-1/Nidogen interacts with four specific immunoglobulin domains of UNC-52 to pattern 2° branches

royalsocietypublishing.org/journal/rsob   Open Biol. 9: 180257

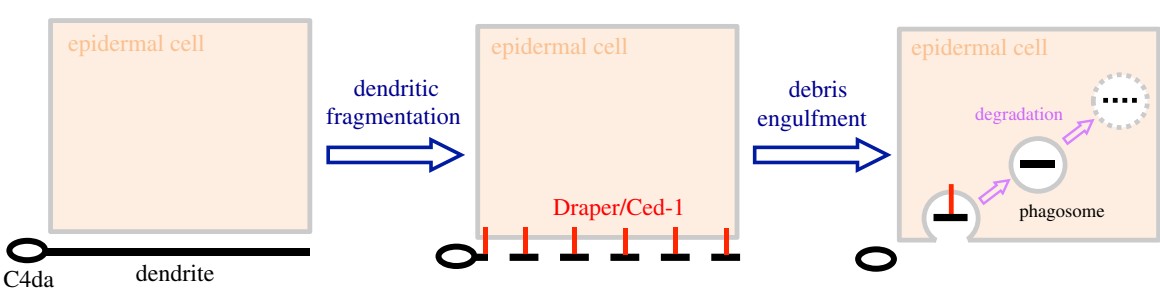

**Figure 3.** Examples of how epidermis shapes dendritic patterns. (*a*) Pre-patterning growth pathways. SAX-7 is expressed in a striped pattern in skin that guides future dendritic patterning before PVD 3° and 4° branch formation. Then, the upcoming 2° branch turns and branches out new branches according to the SAX-7 stripes. (*b*) Synchronizing dendrite growth with body expansion. Initial dendrite growth is rapid until the epidermal field is fully covered at mid-larval stages. This rapid growth of dendrites is decelerated by epidermal *bantam* microRNA to synchronize the growth rate with epidermis for later stages. (*c*) ECM-attached and epidermis-enclosed branches. Dendritic branches are attached on the epidermal ECM by interactions between dendritic integrin (red) and epidermal laminin (green) on the ECM. A portion of an enclosed branch inside an epidermal cell is also shown. The enclosed dendrite is encircled by epidermal membranes (grey) that are invaginated from the basal surface of the epidermal cell. The septate junction protein Coracle (magenta) is expressed in the two closely aligned invaginated membrane layers at the basal side of the enclosed dendrite. (*d*) Cleaning up dendritic debris. During pruning of class IV da (C4da) neurons, dendrites are first fragmented and recognized by the apoptotic cell clearance receptor Draper. Dendritic debris is then engulfed by epidermal cells and further fused to phagosomes for degradation.

[51,52]. However, these UNC-52-NID-1 interactions are dispensable for skin–muscle attachments during 4° branch formation. Therefore, a single ECM molecule can differentially direct dendrite growth in different environments.

The role of muscle-derived UNC-52/Perlecan in PVD branch formation implies that other proximal yet non-contact tissues may modulate dendrite–substrate interactions. Forward genetic screens revealed *lect-2* mutants as exhibiting

disorganized dendrite phenotypes [53,54]. Interestingly, these *lect-2* mutant phenotypes could be rescued by ectopic *lect-2* expression in several different tissues, even though the *lect-2* transcript is localized in muscle cells. Consistent with the idea that LECT-2 is secreted to exert its function in dendritic arborization (figure 2*b*), *lect-2* encodes a homologue of chemotaxin 2 (LECT2)/Chondromodulin II, a chemotactic factor secreted from leucocytes [55,56]. The secreted LECT-2 protein decorates PVD dendritic branches and hypodermal cells in a pattern analogous to SAX-7 expression. As the SAX-7/MNR-1/DMA-1 tripartite complex directs 3° branching, genetic analyses suggest that LECT-2 acts in this pathway by stabilizing ligand–receptor complex formation [57] (figure 2*d*). Thus, the stereotypic menorah-like pattern of PVD dendrites is pre-determined by regulated SAX-7 localization in the hypodermal substrate, thereby guiding dendritic branch growth (figures 2 and 3*a*).

## 3. Stabilizing new branches

Dendritic branch formation necessitates initiation and stabilization of new branches. In *C. elegans*, SAX-7 is pre-patterned in the epidermis, establishing the pathways that guide dendrite growth. As a CAM, it is possible that skin-derived SAX-7 also functions in stabilizing new dendrites. Time-lapse recordings reveal that newly formed PVD 3° branches exhibit dynamic extension and retraction [42]. Whereas more than half of new branches are ultimately stabilized to form T-shape dendrites in wild-type *C. elegans*, nearly all 3° branches in *sax-7* mutants are retracted after branch initiation. Thus, in addition to being a guiding factor, SAX-7 also appears to function as a stabilizer of branches during dendritic growth.

Apart from SAX-7, another skin-derived component, the receptor tyrosine phosphatase CLR-1 was also identified in a *C. elegans* genetic screen [58]. Expression of *clr-1::GFP* was detected in hypodermal and muscle cells but not in PVD neurons. PVD 4° branches are largely disrupted in *clr-1* mutants, but SAX-7 stripes are normal, suggesting an alternative pathway for regulating terminal branch growth. Domain deletion assays on CLR-1 indicate that only the intracellular phosphatase domain, but not the extracellular adhesion domain, is required for branch growth. Genetic analysis also supports that CLR-1 functions parallel to the SAX-7/DMA-1 pathway. Interestingly, unlike SAX-7 that is required both for branch initiation and stabilization, new 4° branches grow onto the SAX-7 stripes of *clr-1* mutants but are ultimately retracted. Hence, hypodermal cells first instruct PVD 4° branch growth through the pre-patterned SAX-7 stripes, and then stabilize those branches through CLR-1 intracellular phosphatase activity.

In *Drosophila* class IV da neurons, numerous dendritic branches that sprout out to innervate the skin (epidermal cells) also require stabilization throughout development. One recent study has shown that mutations in core proteins of HSPG or its biogenesis in epidermal cells strongly impede dendritic growth of class IV da neurons [59]. Time-lapse recordings showed that branches initially grew into HSPG-deficient areas but were not retained there during dynamic growth, suggesting epidermal HSPGs are required for branch stabilization. Consistently, stabilized microtubules were found to be largely depleted from branches located at the borders between wild-type and HSPG-deficient areas.

These data suggest that epidermal HSPGs support dendrite growth by stabilizing and bundling dendritic microtubules. Similar epidermal HSPG-based regulation of dendrite growth has been reported for zebrafish, in which the HSPGs are highly enriched in the basal membranes of the skin [60]. Upon disrupting HSPG synthesis, sensory axons of zebrafish RB neurons failed to reach the skin. Time-lapse examination revealed that epidermal HSPGs in zebrafish guide sensory branches to innervate the skin [60], unlike their role in maintaining branch stability, as is the case for *Drosophila* class IV da neurons.

## 4. Synchronizing dendrite growth with body expansion

During development, dendrites actively extend and branch to cover their target fields; in the case of class IV da neurons, this is the layer of epidermis covering the larval body. As the body size of larvae continuously increases through developmental stages, how is the expansion of the epidermal layer coordinated with dendrite growth to maintain full coverage of the target field? Imaging of early-stage class IV da neurons has revealed that dendrite outgrowth starts around 16 h after egg laying (AEL), after the epidermal sheet has formed [61]. Dendrites rapidly grow and branch to catch up with epidermal sheet expansion until they fully cover the body wall by 48 h AEL (figure 3*b*). Immediately after this rapid growth phase, dendrite growth decelerates to a rate similar to that of epidermal cell expansion (in a process called scaling growth) until pupa formation [61] (figure 3*b*). Interestingly, dendrites were found to greatly interact with epidermal cells at 48 h AEL, coinciding with the growth phase transition [62]. This coincidence implies that epidermal cells may send inhibitory signals to slow down dendrite growth. A genetic screen targeting scaling growth identified the microRNA mutant *bantam*, which exhibits small body size and over-sized dendritic arbors [61]. Although *bantam* is expressed in neurons, epidermal cells and muscle cells, only epidermal expression could rescue the scaling growth defect, suggesting that *bantam* functions non-cell-autonomously in epidermis to inhibit dendrite growth (figure 3*b*). In photo-ablation experiments, da neurons regenerate more dendrites in *bantam* mutants than in wild-type flies, suggesting that *bantam* is involved in generating growth-inhibitory signals in scaling growth [63]. The microRNA *bantam* may restrict dendrite regeneration by downregulating Akt activity that facilitates dendrite regeneration in neurons [63]. In epidermis, *bantam* regulates pathways involved in cell cycle regulation, cell growth and cell adhesion [62]. During development, epidermal cells complete proliferation at embryonic stages and then enter endoreplication around 48 h AEL to scale up cell size and enhance protein synthesis, thereby supporting body expansion [62]. Epithelial endoreplication in larvae also autonomously increases β-integrin expression to enhance interactions with the ECM. In *bantam* mutants, epidermal endoreplication is arrested and results in diverse abnormalities including disrupted dendrite–ECM interactions when dendrites are starting to contact the epidermis extensively. These findings suggest that epidermal cells play active roles in synchronizing dendrite growth when they undergo developmental transitions, thereby matching dendritic arborization to body expansion (figure 3*b*).

royalsocietypublishing.org/journal/rsob    Open Biol. **9**: 180257

royalsocietypublishing.org/journal/rsob   Open Biol. **9**: 180257

## 5. Restricting dendritic branching within a thin two-dimensional space

Maximizing sensory coverage of the innervating field is a fundamental process during polymodal nociceptor morphogenesis. Space-filling neurons such as class IV da neurons may use self-avoidance mechanisms to ensure new branch innervation in regions that are not occupied by other iso-neuronal branches, thereby attaining complete field coverage (figure 1b). Establishing the non-overlapping dendritic pattern of class IV da neurons also requires restricting branch growth in the same plane, which prevents crossovers caused by non-contact passing in three-dimensional space. In fact, these sensory dendrites attach to the epidermis to sense externally derived stimuli. Imaging analyses have revealed that dendritic branches are restricted between the basal surface and ECM of epidermal cells [64,65] (figures 1c and 3c). Dendrite attachment to the ECM is mediated by interactions between integrins in dendrites and epidermis-secreted laminins (figure 3c). In mutants for integrin subunits, many dendritic branches escape ECM attachment and are enclosed by epidermis (figure 3c). Although the functional significance (or defects) associated with dendritic enclosure is not clear, it would allow non-contact crossover of iso-neuronal branches at the basal side of enclosed dendrites. Furthermore, a semaphorin ligand and its receptor are required for dendrite–ECM interactions to avoid crossovers [66]. The epidermis-secreted Sema-2b ligand interacts with the neuronal Plexin B receptor on dendritic surfaces, with this latter also physically associating with integrin subunits. Genetic analyses have indicated that this Sema-2b ligand–receptor complex promotes dendrite positioning at the ECM by activating the downstream Trc kinase. Thus, multiple mechanisms are involved in mediating the ECM–dendrite interaction, ensuring that dendritic branches are positioned in a thin two-dimensional space for executing self-avoidance and maximizing non-redundant dendrite coverage.

Although dendrites are mainly ECM-contacting, some dendrites are enclosed inside epidermal cells in wild-type larvae. A previous study identified that Coracle (Cora), a septate junction protein, is expressed along enclosed dendrites [65], and a follow-up study revealed that Cora is expressed at the basal side of epidermal cells to these enclosed dendrites (figure 3c) [67]. When dendrite positioning was examined, epidermal cora knockdown resulted in less dendritic enclosure, suggesting it is required to enclose dendritic branches [67]. Transmission electron microscopy revealed that enclosed dendrites are encircled by epidermal membranes that are invaginated from the basal surface [64,65], and the two layers of invaginated membranes closely align with each other until they reach the enclosed branch (figure 3c). Cora is a component of septate junction protein complexes, but it is not clear whether other components are also localized and function in dendrite enclosure. Overall, then, epidermal cells actively regulate dendrite positioning at the ECM or their enclosure to establish the non-crossover space-filling pattern.

## 6. Engulfment of fragmented dendrites during pruning

Clearance of degenerating axons and dendrites is an essential process for tissue homeostasis and to avoid inflammation.

Studies of programmed axon pruning in mushroom bodies during metamorphosis and axon degeneration after injury in olfactory receptor neurons indicate that glia, the axon-supporting and -ensheathing cells, mediate clearance of degenerating axons [68,69]. Which cells are involved in dendrite clearance? During Drosophila metamorphosis from larva to adult, dendrites of larval class IV da neurons are pruned completely at the early pupal stage. In these surviving neurons, dendrites are then regenerated from the remaining soma at later pupal stages but with a different pattern in adults. Although early pruning steps, such as severing of dendrites, are largely controlled cell-autonomously, subsequent clearance is conducted on-site by epidermal cells [70]. Similar to how glial cells engulf axonal debris, epidermal cells express the transmembrane protein Draper/Ced-1 to recognize fragmented dendrites (figure 3d). Engulfed dendrite debris is then transported and fused to phagosomes for further degradation [70]. Furthermore, breakdown of long dendrites into short fragments also seems to be regulated by the epidermis, as delayed dendrite fragmentation has frequently been observed upon epidermal knockdown of Draper, Vps16A or WASP to inhibit engulfment or phagosome maturation [71–73]. Time-lapse recordings have revealed that epidermal cells wrap around dendrite-thinning and -beading sites with actin-rich membranes. This actin accumulation was not detected at dendrite–epidermis contact sites when the receptor Draper was depleted by RNAi knockdown [70]. Thus, epidermal cells not only phagocytose dendrite debris, but they also actively promote dendrite fragmentation. The interaction between dendrites and epidermis also extends to lesion-induced dendrite degeneration, which follows the same pathway as the developmentally regulated pruning process [70]. A similar skin-mediated clearance mechanism has also been reported for zebrafish. After laser axotomy of somatosensory peripheral axons in zebrafish, neuronal debris was phagocytosed by epidermal cells rather than by blood-derived phagocytes [74]. Thus, epidermal cells not only act as dendrite supporting and ensheathing cells, but also serve as phagocytes to clean up neuronal debris during programmed or injury-induced pruning.

## 7. Reshaping dendrites

Interestingly, the regenerated dendrites of class IV da neurons further undergo remodelling in early adulthood. At 24 h after eclosion, ventral class IV da neurons rapidly reshape their radially patterned dendrites into a lattice-like pattern that aligns well with lateral tergosternal muscles (LTMs) [75]. This reshaping of dendrites does not involve dendritic pruning and regrowth. Instead, this process is achieved by relocating the existing branches to the grooves between LTM fibres [75]. By a genetic screen, mutations in matrix metalloproteinase 2 (Mmp2) were found to cause obvious defects in dendrite reshaping [75]. Mmp is known to modify the basement membrane by degrading ECM components [76]. Consistent with its molecular functions, Mmp2 expression was found to be transiently elevated in epidermal cells when the ECM between the epidermis and the LTMs is being selectively degraded [75]. These findings suggest that epidermis actively modifies the ECM microenvironment to allow relocation of dendritic branches.

# 8. Concluding remarks and future prospects

Using powerful genetic tools to manipulate gene expression, accumulating evidence from studies of *C. elegans* and *Drosophila* has shed light on the active roles of epidermis in regulating dendrite patterning and degeneration. Epidermal cells, the substrate of sensory dendrites, instruct and stabilize dendritic growth by forming SAX-7/L1CAM stripes in *C. elegans*. L1CAM family proteins are conserved and required for neural development in other organisms like *Drosophila* and mice. It is possible that substrates in higher organisms may also express L1CAM homologues to regulate dendritic growth. Although only one isoform of *sax-7* is essential for dendritic growth in nematodes [43], both fly and mouse express alternatively spliced isoforms in neurons and non-neuronal tissues [44]. Thus, substrates could employ these alternatively spliced isoforms to regulate dendrite arborization. Although dynein components and fusogen are required for SAX-7 expression into stripes, it is unclear how these SAX-7 stripes are localized along the muscle–skin interface. It is possible that muscles express unidentified cues that recruit and stabilize hypodermal SAX-7 expression.

In *Drosophila*, the dendrites of da neurons are restricted to a thin space between the basal surface of epidermis and the ECM. Through self-avoidance, dendritic arbors do not overlap. However, some dendrites are enclosed by epidermal cells [64,65]. The percentage of enclosed dendrites progressively increases from second instar to the final larval stages [62], suggesting an increasing requirement in later stages. Although dendritic ensheathment could allow other branches to crossover without direct contact, some enclosed dendrites are solitary without crossover, suggesting unidentified functions for enclosed dendrites [66]. Interestingly, damaged epidermal cells are reported to secrete an inflammatory cytokine Eiger, a homologue of tumour necrosis factor, to induce thermal hyperalgesia and allodynia through da neurons, suggesting that epidermal cells could also play a role in transmitting signals to neurons [77].

Even though epidermal cells support dendritic arborization during larval stages, they promote dendrite degeneration during pupal stages, suggesting an interesting transition from protective to attacking roles. Larval epidermal cells are also degenerated during pupal stages to be replaced by adult epidermal cells. During larval stages, lesion-induced dendritic pruning also requires epidermal cells to exert an attacking role. Epidermal cells seem to be able to interchange these two conflicting roles at any time, depending on the status of dendrites. The lessons learned from invertebrate studies could potentially be extrapolated, with some variations, to mammalian systems, with considerable implications for sensory regeneration in patients suffering burns or physical lesions.

Data accessibility. This article has no additional data.

Competing interests. We declare we have no competing interests.

Funding. We received no funding for this study.

Acknowledgements. We thank Chun-Liang Pan at National Taiwan University for comments on the manuscript, as well as Chan-Yen Ou and Ying-Chun Chen at National Taiwan University for providing the image for figure 2a.

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
