## [Reviewer comments · Open Biology]

Review History

RSOB-18-0257.R0 (Original submission)

Review form: Reviewer 1

Recommendation

Major revision is needed (please make suggestions in comments)

Are each of the following suitable for general readers?

- a) **Title**
Yes
- b) **Summary**
Yes
- c) **Introduction**
No

Is the length of the paper justified?

Yes

Should the paper be seen by a specialist statistical reviewer?

No

Is it clear how to make all supporting data available?

Not Applicable

Is the supplementary material necessary; and if so is it adequate and clear?

Not Applicable

Do you have any ethical concerns with this paper?

No

Comments to the Author

In both invertebrates and vertebrates, animal skins are innervated by sensory neurons that allow animals to sense and respond to the external world. Although the sensory neurites closely associate with epidermal cells, the interactions between neurites and epidermal cells have not been carefully characterized until recently. In the manuscript by Yang and Chien, the authors attempt to summarize our current knowledge on these interactions, by primarily focusing on work done in *C. elegans* and *Drosophila*. Although there are already good reviews that cover similar subjects (e.g. Dong et al, *Annu Rev Physiol* 2015), this manuscript discusses some of the more recent findings. In this regard, the review could be a welcomed addition to the literature.

On the other hand, in my opinion, the manuscript suffers from several important deficits. I would only be supportive for the publication of this review in *Open Biology* if these issues are satisfyingly addressed.

Major comments:

1. The scope of the review is kind of narrow in my opinion. There is a larger body of work on this topic in recent literature that the authors seemed to have ignored. First, nice work has been done in vertebrate systems like zebrafish to investigate sensory neurite/epidermal cell interactions (Wang et al, *Current Biology*, 2012, DOI 10.1016/j.cub.2012.01.040; Rasmussen et al, *J Neurosci*, 2015, DOI:10.1523/JNEUROSCI.3613-14.2015). I understand that the authors chose to focus on work done in worms and flies, but the interactions in the zebrafish are very similar to the those discussed in the manuscript and have added to our understanding of neurite/epidermis interactions as a whole. These should at least be acknowledged. Second, even in the *Drosophila* literature, there are important work missed by the manuscript. Some examples are DOI 10.1016/j.devcel.2010.02.010; DOI 10.1016/j.cub.2009.03.062; DOI 10.1016/j.celrep.2017.09.001. The findings reported in these studies should be discussed.
2. The structure of this manuscript is awkward. It starts by describing the PVD system in *C. elegans*, which is followed by two sections of how PVD dendrites are affected by the hypoderm. But the *Drosophila* da neuron system was introduced as a part of the second section, where the introduction took a large space. The relevant *Drosophila* work for this section is only a smaller part at the end of the second section. The rest of the manuscript is then focused on *Drosophila* work only. This organization makes the manuscript hard to follow. I suggest introducing both systems in the beginning and then grouping worm and fly work under appropriate sections. There should also be discussions on how similar and how different the fly and worm systems are.
3. The writing in many places lacks clarity or is confusing. This is especially true in later part of the manuscript when fly work was discussed. Those places could probably be understood only by experts working on these specific systems. However, the purpose of writing a review is to

inform non-experts on specific topics. I suggest the authors to carefully revise the text so that scientists who are not in the field can also understand it.

Here are just a few examples of the confusing writing. On page 9, "This self-avoidance mechanism simply maximizes the branching coverage without needing of directing every single dendrite growth." This sentence is very hard to understand. In the next sentence "Although the lumen between the muscle and epidermis is spacious...", I don't think the space between the muscle and the epidermis can be called a lumen because muscles and the epidermis do not form a concealed compartment. The next sentence, "Indeed, confocal imaging reveals the dorsal field of a *Drosophila* larvae hemi-segment to be covered by hundreds of dendritic branches distributed within a narrow space". It would be very hard for non-experts to imagine what is the narrow space the authors refer to. On page 10, "the Semaphorin ligand and receptor" is confusing. Probably "a semaphorin ligand and its receptor" is more accurate.

On page 10 in the section of "Fragmented dendrite engulfment during dendrite pruning", the sentence "Studies of axon pruning..." lacks the necessary contexts for people to understand it. What axons? Which systems?

4. The figures are not as helpful as they could be. Since the review is about interactions between sensory dendrites and epidermal cells, diagrams illustrating the spatial organization of these two types of tissues in *C. elegans* and *Drosophila* are necessary. Also it would be helpful to include figures summarizing the major ways epidermal cells regulate dendrite morphogenesis.

5. Lastly, I think the English writing of the manuscript should be improved. It may be appropriate to seek help from English editing service.

Minor comments:

1. In the last sentence of the abstract, "myriad" is not accurate because it means "numerous". But in fact the paper only talked about five roles of epidermis.

2. I am not sure if the *C. elegans* PVD neurons can be considered as space-filling neurons and I have not seen them referred to as such in the literature. Unlike space-filling neurons in the fly and the zebrafish, PVD dendrites do not fully cover the body surface.

3. In the first paragraph, the claim of "mainly through studies of *Drosophila* and *C. elegans*" is not correct. Important work on the interactions between sensory neurites and the epidermis has also been done Zebrafish.

4. On page 8, the authors mentioned that class IV da neurons appear around 16 hr AEL, but in the literature the earliest time these neurons were observed is in stage-16 embryos at around 13 hr AEL (Grueber et al, *Current Biology*, 2003; Han et al; *PNAS*, 2011).

Decision letter (RSOB-18-0257.R0)

28-Jan-2019

Dear Professor Chien,

We are pleased to inform you that your manuscript RSOB-18-0257 entitled "Beyond being innervated: the epidermis actively shapes sensory dendritic patterning" has been accepted by the

Editor for publication in Open Biology. The reviewer(s) have recommended publication, but also suggest some minor revisions to your manuscript. Therefore, we invite you to respond to the reviewer(s)' comments and revise your manuscript.

Please submit the revised version of your manuscript within 14 days. If you do not think you will be able to meet this date please let us know immediately and we can extend this deadline for you.

- 1) A text file of the manuscript (doc, txt, rtf or tex), including the references, tables (including captions) and figure captions. Please remove any tracked changes from the text before submission. PDF files are not an accepted format for the "Main Document".
- 2) A separate electronic file of each figure (tiff, EPS or print-quality PDF preferred). The format should be produced directly from original creation package, or original software format. Please note that PowerPoint files are not accepted.
- 3) Electronic supplementary material: this should be contained in a separate file from the main text and meet our ESM criteria (see <http://royalsocietypublishing.org/instructions-authors#question5>). All supplementary materials accompanying an accepted article will be treated as in their final form. They will be published alongside the paper on the journal website and posted on the online figshare repository. Files on figshare will be made available approximately one week before the accompanying article so that the supplementary material can be attributed a unique DOI.

Online supplementary material will also carry the title and description provided during submission, so please ensure these are accurate and informative. Note that the Royal Society will not edit or typeset supplementary material and it will be hosted as provided. Please ensure that the supplementary material includes the paper details (authors, title, journal name, article DOI). Your article DOI will be 10.1098/rsob.2016[last 4 digits of e.g. 10.1098/rsob.20160049].

- 4) A media summary: a short non-technical summary (up to 100 words) of the key findings/importance of your manuscript. Please try to write in simple English, avoid jargon, explain the importance of the topic, outline the main implications and describe why this topic is newsworthy.

Images

Data-Sharing

It is a condition of publication that data supporting your paper are made available. Data should be made available either in the electronic supplementary material or through an appropriate repository. Details of how to access data should be included in your paper. Please see <http://royalsocietypublishing.org/site/authors/policy.xhtml#question6> for more details.

Data accessibility section

Sincerely,
The Open Biology Team
<mailto:openbiology@royalsociety.org>

Reviewer(s)' Comments to Author:

Referee:

Comments to the Author(s)

In both invertebrates and vertebrates, animal skins are innervated by sensory neurons that allow animals to sense and respond to the external world. Although the sensory neurites closely associate with epidermal cells, the interactions between neurites and epidermal cells have not been carefully characterized until recently. In the manuscript by Yang and Chien, the authors attempt to summarize our current knowledge on these interactions, by primarily focusing on work done in *C. elegans* and *Drosophila*. Although there are already good reviews that cover similar subjects (e.g. Dong et al, *Annu Rev Physiol* 2015), this manuscript discusses some of the more recent findings. In this regard, the review could be a welcomed addition to the literature.

On the other hand, in my opinion, the manuscript suffers from several important deficits. I would only be supportive for the publication of this review in Open Biology if these issues are satisfyingly addressed.

Major comments:

1. The scope of the review is kind of narrow in my opinion. There is a larger body of work on this topic in recent literature that the authors seemed to have ignored. First, nice work has been done in vertebrate systems like zebrafish to investigate sensory neurite/epidermal cell interactions (Wang et al, *Current Biology*, 2012, DOI 10.1016/j.cub.2012.01.040; Rasmussen et al, *J Neurosci*,

2015, DOI:10.1523/JNEUROSCI.3613-14.2015). I understand that the authors chose to focus on work done in worms and flies, but the interactions in the zebrafish are very similar to the those discussed in the manuscript and have added to our understanding of neurite/epidermis interactions as a whole. These should at least be acknowledged. Second, even in the *Drosophila* literature, there are important work missed by the manuscript. Some examples are DOI 10.1016/j.devcel.2010.02.010; DOI 10.1016/j.cub.2009.03.062; DOI 10.1016/j.celrep.2017.09.001. The findings reported in these studies should be discussed.

2. The structure of this manuscript is awkward. It starts by describing the PVD system in *C. elegans*, which is followed by two sections of how PVD dendrites are affected by the hypoderm. But the *Drosophila* da neuron system was introduced as a part of the second section, where the introduction took a large space. The relevant *Drosophila* work for this section is only a smaller part at the end of the second section. The rest of the manuscript is then focused on *Drosophila* work only. This organization makes the manuscript hard to follow. I suggest introducing both systems in the beginning and then grouping worm and fly work under appropriate sections. There should also be discussions on how similar and how different the fly and worm systems are.

3. The writing in many places lacks clarity or is confusing. This is especially true in later part of the manuscript when fly work was discussed. Those places could probably be understood only by experts working on these specific systems. However, the purpose of writing a review is to inform non-experts on specific topics. I suggest the authors to carefully revise the text so that scientists who are not in the field can also understand it.

Here are just a few examples of the confusing writing. On page 9, "This self-avoidance mechanism simply maximizes the branching coverage without needing of directing every single dendrite growth." This sentence is very hard to understand. In the next sentence "Although the lumen between the muscle and epidermis is spacious...", I don't think the space between the muscle and the epidermis can be called a lumen because muscles and the epidermis do not form a concealed compartment. The next sentence, "Indeed, confocal imaging reveals the dorsal field of a *Drosophila* larvae hemi-segment to be covered by hundreds of dendritic branches distributed within a narrow space". It would be very hard for non-experts to imagine what is the narrow space the authors refer to. On page 10, "the Semaphorin ligand and receptor" is confusing. Probably "a semaphorin ligand and its receptor" is more accurate.

On page 10 in the section of "Fragmented dendrite engulfment during dendrite pruning", the sentence "Studies of axon pruning..." lacks the necessary contexts for people to understand it. What axons? Which systems?

4. The figures are not as helpful as they could be. Since the review is about interactions between sensory dendrites and epidermal cells, diagrams illustrating the spatial organization of these two types of tissues in *C. elegans* and *Drosophila* are necessary. Also it would be helpful to include figures summarizing the major ways epidermal cells regulate dendrite morphogenesis.

5. Lastly, I think the English writing of the manuscript should be improved. It may be appropriate to seek help from English editing service.

Minor comments:

1. In the last sentence of the abstract, "myriad" is not accurate because it means "numerous". But in fact the paper only talked about five roles of epidermis.

2. I am not sure if the *C. elegans* PVD neurons can be considered as space-filling neurons and I have not seen them referred to as such in the literature. Unlike space-filling neurons in the fly and the zebrafish, PVD dendrites do not fully cover the body surface.

3. In the first paragraph, the claim of “mainly through studies of *Drosophila* and *C. elegans*” is not correct. Important work on the interactions between sensory neurites and the epidermis has also been done Zebrafish.

4. On page 8, the authors mentioned that class IV da neurons appear around 16 hr AEL, but in the literature the earliest time these neurons were observed is in stage-16 embryos at around 13 hr AEL (Grueber et al, *Current Biology*, 2003; Han et al; *PNAS*, 2011).

Author's Response to Decision Letter for (RSOB-180257.R0)

See Appendix A.

Decision letter (RSOB-18-0257.R1)

04-Mar-2019

Dear Professor Chien

We are pleased to inform you that your manuscript entitled "Beyond being innervated: the epidermis actively shapes sensory dendritic patterning" has been accepted by the Editor for publication in *Open Biology*.

Thank you for your fine contribution. On behalf of the Editors of *Open Biology*, we look forward to your continued contributions to the journal.

Sincerely,

The Open Biology Team
mailto:openbiology@royalsociety.org

Appendix A

Reviewer(s)' Comments to Author:

Referee:

Comments to the Author(s)

In both invertebrates and vertebrates, animal skins are innervated by sensory neurons that allow animals to sense and respond to the external world. Although the sensory neurites closely associate with epidermal cells, the interactions between neurites and epidermal cells have not been carefully characterized until recently. In the manuscript by Yang and Chien, the authors attempt to summarize our current knowledge on these interactions, by primarily focusing on work done in *C. elegans* and *Drosophila*. Although there are already good reviews that cover similar subjects (e.g. Dong et al, *Annu Rev Physiol* 2015), this manuscript discusses some of the more recent findings. In this regard, the review could be a welcomed addition to the literature.

On the other hand, in my opinion, the manuscript suffers from several important deficits. I would only be supportive for the publication of this review in *Open Biology* if these issues are satisfyingly addressed.

Major comments:

1. The scope of the review is kind of narrow in my opinion. There is a larger body of work on this topic in recent literature that the authors seemed to have ignored. First, nice work has been done in vertebrate systems like zebrafish to investigate sensory neurite/epidermal cell interactions (Wang et al, *Current Biology*, 2012, DOI 10.1016/j.cub.2012.01.040; Rasmussen et al, *J Neurosci*, 2015,

DOI:10.1523/JNEUROSCI.3613-14.2015). I understand that the authors chose to focus on work done in worms and flies, but the interactions in the zebrafish are very similar to the those discussed in the manuscript and have added to our understanding of neurite/epidermis interactions as a whole. These should at least be acknowledged. Second, even in the *Drosophila* literature, there are important work missed by the manuscript. Some examples are DOI 10.1016/j.devcel.2010.02.010; DOI 10.1016/j.cub.2009.03.062; DOI 10.1016/j.celrep.2017.09.001. The findings reported in these studies should be discussed.

We are grateful to this reminding.

(1) The zebrafish works have been cited as well as discussed and in the ending of “Stabilizing new branches” section in page 8, and in the section of “Engulfment of fragmented dendrites during pruning” in the top of page 12.

(2) The fly works have also been cited and discussed as:

(a) DOI 10.1016/j.devcel.2010.02.010 is in the section of “Reshaping dendrites” in page 12-13.

(b) DOI 10.1016/j.cub.2009.03.062 is in the section of “Concluding remarks and future prospects” in page 13.

(c) DOI 10.1016/j.celrep.2017.09.001 is in the section of “Restricting dendritic branching within a thin two-dimensional space” in page 11.

2. The structure of this manuscript is awkward. It starts by describing the PVD system in *C. elegans*, which is followed by two sections of how PVD dendrites are affected by the hypoderm. But the *Drosophila* da neuron system was introduced as a part of the second section, where the introduction took a large space. The relevant *Drosophila* work for this section is only a smaller part at the end of the second section. The rest of the manuscript is then focused on *Drosophila* work only. This organization makes the manuscript hard to follow. I suggest introducing both systems in the beginning and then grouping worm and fly work under appropriate

sections. There should also be discussions on how similar and how different the fly and worm systems are.

We are grateful for reviewer to point out the weakness of our review structure. The introduction of both systems is reorganized into the beginning and also includes a simple description of the somatosensory RB neurons in zebrafish. The differences are discussed in the end paragraph of the introduction.

3. The writing in many places lacks clarity or is confusing. This is especially true in later part of the manuscript when fly work was discussed. Those places could probably be understood only by experts working on these specific systems. However, the purpose of writing a review is to inform non-experts on specific topics. I suggest the authors to carefully revise the text so that scientists who are not in the field can also understand it.

Thanks for reminding, we have carefully and largely revised the part of fly works by expressing ideas and writing sentences clearer as well as easier for reading by outfield readers.

Here are just a few examples of the confusing writing. On page 9, “This self-avoidance mechanism simply maximizes the branching coverage without needing of directing every single dendrite growth.” This sentence is very hard to understand. In the next sentence “Although the lumen between the muscle and epidermis is spacious...”, I don’t think the space between the muscle and the epidermis can be called a lumen because muscles and the epidermis do not form a concealed compartment. The next sentence, “Indeed, confocal imaging reveals the dorsal field of a *Drosophila* larvae hemi-segment to be covered by hundreds of dendritic branches distributed within a narrow space”. It would be very hard for non-experts to imagine what is the narrow space the authors refer to. On page 10,

“the Semaphorin ligand and receptor” is confusing. Probably “a semaphorin ligand and its receptor” is more accurate.

On page 10 in the section of “Fragmented dendrite engulfment during dendrite pruning”, the sentence “Studies of axon pruning...” lacks the necessary contexts for people to understand it. What axons? Which systems?

For these specific examples. We have modified the text as below:

(1) The self-avoidance is introduced more in the end of introduction section when mentioning the differences between class IV da and PVD neurons, through page 4 line 29 to page 5 line 2.

(2) The whole section of “Restricting dendritic branching within a thin two-dimensional space” is largely rewritten to avoid confusing sentences and words.

(3) The sentence in the section of “Engulfment of fragmented dendrites during pruning” is rewritten as “Studies of programmed axon pruning in mushroom bodies during metamorphosis and axon degeneration after injury in olfactory receptor neurons indicate that glia, the axon-supporting and -ensheathing cells, mediate clearance of degenerating axons.” in page 11 lines 26-29.

4. The figures are not as helpful as they could be. Since the review is about interactions between sensory dendrites and epidermal cells, diagrams illustrating the spatial organization of these two types of tissues in *C. elegans* and *Drosophila* are necessary. Also it would be helpful to include figures summarizing the major ways epidermal cells regulate dendrite morphogenesis.

We are grateful for reviewer to give us constructive suggestions. We have included additional figures to illustrate the spatial organization of class IV da/PVD neurons to

the epidermis in Figure 1 and 2. Also, cartoons that describing examples of how epidermis can regulate dendrite patterning are added in Figure 3.

5. Lastly, I think the English writing of the manuscript should be improved. It may be appropriate to seek help from English editing service.

We have sent the revision version for English editing.

Minor comments:

1. In the last sentence of the abstract, “myriad” is not accurate because it means “numerous”. But in fact the paper only talked about five roles of epidermis.

Thanks for reminding, we have changed it into “several aspects” in page 1, line 20.

2. I am not sure if the *C. elegans* PVD neurons can be considered as space-filling neurons and I have not seen them referred to as such in the literature. Unlike space-filling neurons in the fly and the zebrafish, PVD dendrites do not fully cover the body surface.

Thanks for reminding, we have removed the description of PVD as a space-filling neuron in the introduction.

3. In the first paragraph, the claim of “mainly through studies of *Drosophila* and *C. elegans*” is not correct. Important work on the interactions between sensory neurites and the epidermis has also been done Zebrafish.

Thanks for reminding, we have added zebrafish into this sentence and cited a review for sensory neurites- epidermis interactions in zebrafish in page 2, lines 13-18.

4. On page 8, the authors mentioned that class IV da neurons appear around 16 hr AEL, but in the literature the earliest time these neurons were observed is in stage-16 embryos at around 13 hr AEL (Grueber et al, Current Biology, 2003; Han et al; PNAS, 2011).

Thanks for reminding, we have reorganized the sentence as “Imaging of early-stage class IV da neurons has revealed that dendrite outgrowth starts around 16 hr after egg laying (AEL), after the epidermal sheet has formed (Parrish, et al., 2009).” in page 9, lines 18-20.